# Epidemiology of Meningococcal Disease in Four South American Countries and Rationale of Vaccination in Adolescents from the Region: Position Paper of the Latin American Society of Pediatric Infectious Diseases (SLIPE)

**DOI:** 10.3390/vaccines11121841

**Published:** 2023-12-12

**Authors:** Rodolfo Villena, Marco Aurelio Safadi, Ángela Gentile, Mónica Pujadas, Verónica De la Maza, Sergio George, Juan Pablo Torres

**Affiliations:** 1Department of Pediatrics, Hospital de Niños Exequiel González Cortés, Faculty of Medicine, Universidad de Chile, Santiago 8900085, Chile; rvillena@uchile.cl; 2Department of Pediatrics, School of Medical Sciences, Santa Casa de Sao Paulo, Sao Paulo 01224-001, Brazil; masafadi@uol.com.br; 3Department of Epidemiology, Hospital de Niños Ricardo Gutierrez, Faculty of Medicine, Universidad de Buenos Aires, Ciudad Autónoma de Buenos Aires C1121, Argentina; angelagentile@fibertel.com.ar; 4Department of Epidemiology and Pediatrics Infectious Diseases, Hospital Pereira Rossell, Faculty of Medicine, University of the Republic, Montevideo 11400, Uruguay; monipujadas@gmail.com; 5Department of Pediatrics, Hospital Dr. Luis Calvo Mackenna, Faculty of Medicine, Universidad de Chile, Santiago 7500539, Chile; vdelamaza@uchile.cl (V.D.l.M.); sgeorge@ug.uchile.cl (S.G.)

**Keywords:** *Neisseria meningitidis*, meningococcal disease, adolescents, epidemiology, vaccines, South America

## Abstract

Surveillance of meningococcal disease (MD) is crucial after the implementation of vaccination strategies to monitor their impact on disease burden. Adolescent vaccination could provide direct and indirect protection. Argentina, Brazil, and Chile have introduced meningococcal conjugate vaccines (MCV) into their National Immunization Programs (NIP), while Uruguay has not. Here, we analyze the epidemiology of MD and vaccination experience from these four South American countries to identify needs and plans to improve the current vaccination programs. Methodology: Descriptive study of MD incidence rates, serogroup distribution, case fatality rates (CFR), and MCV uptakes during the period 2010–2021 in Argentina, Brazil, Chile, and Uruguay. Data were extracted from national surveillance programs, reference laboratories, NIPs, and Pubmed. Results: MD overall incidence from 2010 to 2021 have a decreasing trend in Argentina (0.37 [IQR = 0.20–0.61]), Brazil (0.59 [IQR = 0.54–1.22]), and Chile (0.45 [IQR = 0.40–0.77]), while a significant increase in Uruguay (0.47 [IQR = 0.33–0.69]) was found from 2016 to 2019. During the COVID-19 pandemic, all countries sharply reduced their MD incidence. The highest incidence rates were observed among infants, followed by children 1–4 years of age. No second peak was evident in adolescents. A reduction in serogroup C, W, and Y cases has occurred in Argentina, Brazil, and Chile after introduction of MCV, serogroup B becoming predominant in all four countries. Median CFR was 9.0%, 21%, 19.9%, and 17.9% in Argentina, Brazil, Chile, and Uruguay, respectively. Median uptake of MCV for Argentina and Brazil were 66.6% and 91.0% for priming in infants; 54.7% and 84.5% for booster in toddlers; and 47.5% and 53% for adolescents; while for Chile, 95.6% for toddlers. Conclusions: Experience after the implementation of MCV programs in South America was successful, reducing the burden of MD due to the vaccine serogroups. High vaccine uptake and the inclusion of adolescents will be crucial in the post-pandemic period to maintain the protection of the population. The increase in the proportion of serogroup B cases emphasizes the importance of continuous surveillance to guide future vaccination strategies.

## 1. Introduction

Meningococcal disease is a public health problem, mainly caused worldwide by serogroups A, B, C, W, Y, and X of *Neisseria meningitidis* [1]. Its dynamic and unpredictable epidemiology challenge the strategies to prevent or predict its occurrence [2]. Children below 5 years of age have the highest MD incidence rates globally [3], and the risk of MD and hospitalization is almost three times higher in infants <12 months of age compared to children between 1 and 4 years of age [4]. A second peak in incidence is seen in adolescents and young [3] or older adults [5], depending on the country, year, and serogroup. Pharyngeal carriage in adolescents is considered the main reservoir for *N. meningitidis* and the source of transmission to other susceptible age groups [6]. Asymptomatic carriage increases through childhood, reaching a peak in adolescents and young adults, subsequently decreasing in older ages [7]. 

In addition to natural fluctuation, a global decrease in MD trends has been seen during the past 20 years, mostly influenced by vaccination programs [8]. Monovalent A (MCV-A) and C (MCV-C), as well as quadrivalent ACWY meningococcal conjugated vaccines (MCV-ACWY) proved to be highly effective [9,10,11], while two recombinant protein vaccines have been approved to protect against serogroup B (MenB) [12,13,14]. Most countries initially focused vaccine strategies on infants. However an increasing number of countries from Europe, North America, and Latin America are routinely vaccinating adolescents and young adults [6,15,16] based on evidence of the effectiveness of conjugated meningococcal vaccines in preventing acquisition of carriage in this age group [17,18,19], achieving indirect protection to unvaccinated cohorts. In this direction, the first pentavalent ABCWY vaccine has been recently approved by FDA in USA for use in adolescents and young adults [20]. 

The World Health Organization (WHO) has launched a global roadmap to defeat meningitis as a public health threat by 2030 [21,22], which requires to combine multiple efforts that include prevention and epidemic control, including enhancement of accessibility to vaccines; improved diagnosis and treatment; strong national surveillance systems to document the effect of vaccines and the burden of disease; effective support and care for people affected by meningitis; and advocacy and engagement to raise awareness of meningitis and its impact to achieve its main objectives, and thus to eliminate epidemics of bacterial meningitis, reduce cases of vaccine-preventable bacterial meningitis by 50% and deaths by 70%, and finally to reduce disability and improve quality of life after meningitis of any cause. Still, MD is probably underreported in Latin America, but efforts have been made to improve surveillance systems [23]. A meta-analysis published in 2017 analyzed the burden of MD in Latin America, which focused on the period 2008–2018, and most of the studies were published before 2017 [24,25]. After this, reports about national MD epidemiology only from Argentina [26], Brazil [24,27,28] and Chile [10] have been published, but there are no recent comparative analyses between different countries.

The purpose of this manuscript is to review the epidemiology of MD in four South American countries (Argentina, Brazil, Chile, and Uruguay) with well-established surveillance systems during the period 2010–2021 and lessons learned from the countries with MCV programs already implemented, as well as discuss future strategies that could optimize the impact of these programs on the burden of MD. 

## 2. Materials and Methods

### 2.1. Overall Design

A descriptive analysis of incidence and case fatality rate (CFR) of reported MD cases in Argentina, Brazil, Chile, and Uruguay between 2010 and 2021 was performed. Epidemiologic data were obtained from the Ministry of Health (MoH) of Argentina, Brazil, Chile, and Uruguay databases, based on mandatory reporting of suspect MD cases to the respective MoH Epidemiology Department for each country. Microbiological diagnosis of *Neisseria meningitidis* was performed based on positive cultures or polymerase chain reaction (PCR) from sterile sites. Serogroup classification of confirmed samples was performed using agglutination with polyclonal antibodies and/or PCR at the National Laboratories of reference for *N meningitidis* for each country. Serogroup distribution data from Argentina were collected from the SIREVA database [29]. Meningococcal isolates belonging to the same MD case accounted for one case. Pharyngeal carriage rates were extracted from published manuscripts available in the Pubmed database during the last 13 years. 

### 2.2. Statistical Analysis

MD overall incidence rates per 100,000 population were calculated with numerators based on the MD case numbers and denominators using age-specific population data. CFRs were expressed as percentages of total reported confirmed MD cases, and serogroup-specific distributions were expressed as percentages of total laboratory-confirmed MD cases. Vaccine uptake rates were reported as percentages of total age-specific population. Continuous variables were reported as medians and interquartile range (IQR), calculated with GraphPad Prism version 9.0.1.

## 3. Results

### 3.1. MD in Argentina 

#### 3.1.1. Incidence

Median overall incidence rate during the period 2010–2021 in Argentina was 0.37/100,000 (IQR = 0.20–0.61), with a sustained decrease from 2013 to 2021, mainly from 2020 (Figure 1). When data corresponding to years 2020 and 2021 were excluded, median incidence rate in Argentina increased to 0.41/100,000 (IQR = 0.31–0.62) (Figure 1). Percentage decrease of overall incidence in 2020 compared to the median in the period 2010–2019 was 87.9% (Table 1). MD incidence depicted by age and year was not available (Figure 2). 

#### 3.1.2. Serogroup Distribution

Serogroup W (MenW) was slightly predominant since 2010 to 2012 in Argentina (ranging from 48% in 2011 to 55.8% in 2012), while MenB increased to comparable rates in 2013 and 2014 and became predominant from 2015 to 2021 (ranging from 46% in 2019 to 70% in 2020). No cases due to serogroup A (MenA) were reported (Figure 3). 

#### 3.1.3. Case Fatality Rates

Median CFR during the period 2010–2019 was 9.0% (IQR = 6.8–11.8%), with a peak of 17% in 2018 (Figure 4). 

#### 3.1.4. Carriage

The oropharyngeal carriage study performed in children from 1 to 17 years old attending to one hospital in Buenos Aires, Argentina [30], showed an overall 6.5% for meningococcal carriage, but depicted by age the results were higher in the adolescent’s cohorts from 10–17 years, reaching 9.4%, mainly associated with passive smoking and pub/nightclub visits. 

#### 3.1.5. Vaccine Strategy and Uptake

A routine vaccination program against *N. meningitidis* in infants and adolescents has been implemented in Argentina (Table 2). Median uptake was 77.4% (IQR = 75.5–80.7%) at 3 months of age, decreased at older ages to a median of 66.6% (IQR = 58.5–74.7%) at 5 months of age (during the period 2017–2021), 54.7% (IQR = 45.4–74.1%) at 15 months of age (2018–2021), and 47.5% (IQR = 31–60%) at 11 years of age (2017–2021) (Table 2, and Appendix A). 

### 3.2. MD in Brazil 

#### 3.2.1. Incidence

Median overall incidence during the period 2010–2021 was 0.59/100,000 in Brazil (IQR = 0.54–1.22), with a sustained decrease observed until 2016 reaching a plateau from 2016 to 2019 (Figure 1). 94.5%. When data corresponding to pandemic period were excluded, median incidence rate increased to 0.72/100,000 (IQR = 0.55–1.32), with a percentage decrease of 76.3% in 2020 compared to the median in the period 2010–2019 (Table 1). 

#### 3.2.2. Serogroup Distribution

Serogroup C (MenC) remained the predominant serogroup causing MD during the period 2010–2020, although there was a progressive decrease (from 80% in 2010 to 47% in 2020) since the introduction of the MCV-C program in infants in 2010 and then the introduction of the MCV-C booster dose in adolescents in 2017, with a consequent increase in the proportion of cases caused by MenB, which became the predominant serogroup in 2021 responsible for 50% of MD cases (Figure 2). No cases due to serogroup A (MenA) were reported. 

#### 3.2.3. Case Fatality Rates

Median CFR during the period 2010–2021 was 21.0% (IQR = 20.2–22%), with a peak of 23% in 2017 (Figure 4). 

#### 3.2.4. Carriage

Two studies evaluating pharyngeal carriage in Brazil were performed [31,32]: one included 1208 students 11–19 years of age in Campinas, finding an overall carriage prevalence of 9.9%, and the second, performed in healthy subjects aged 1–24 years in Embu das Artes city, São Paulo, found the highest carriage prevalence in adolescents 10–19 years old, reaching 12.5%. 

#### 3.2.5. Vaccine Strategy and Uptake

Brazil introduced MCV against MenC in infants in 2010, being the first of the analyzed countries to include a vaccine against *N. meningitidis* in its NPI (Table 2). In 2017, routine vaccination against MenC was implemented in adolescents, replaced in 2020 by tetravalent MCV-ACWY. Vaccine uptake during the evaluated period was higher in infants and toddlers, with a median of 91.0% (IQR = 87–98%) at 3–5 months (2011–2021) and 84.5% (IQR = 77.2–88.5%) at 12–15 months of age (2012–2021), but decreased to 53% (IQR = 43.5–68.5%) at 11–12 years of age (2018–2021) (Table 2, and Appendix A). 

### 3.3. MD in Chile 

#### 3.3.1. Incidence

Median overall incidence during the period 2010–2021 was 0.45/100,000 (IQR = 0.40–0.77), with a peak of 0.80/100,000 from 2012 to 2014 and a posterior sustained decrease from 2014 to 2018. When data corresponding to pandemic period were excluded, median incidence rate increased to 0.55/100,000 (IQR = 0.40–0.80), with a percentual decrease of 94.5% in 2020 compared to the median in the period 2010–2019 (Table 1). 

#### 3.3.2. Serogroup Distribution

MenB was the predominant serogroup causing MD in 2010–2011, replaced by MenW since 2012, reaching a peak of 68% of MD cases in 2014. A progressive decrease of MenW was seen from 2014 to 2021, with a consequent increase in the proportion of cases due to MenB, which reached a peak of 66% of MD cases in 2020. No cases due to MenA were reported during the period (Figure 3).

#### 3.3.3. Case Fatality Rates

Median CFR during the period 2010–2021 was 19.9% (IQR = 15.3–27.8%) with a peak of 30% in 2017 (Figure 4).

#### 3.3.4. Carriage

Two studies evaluating pharyngeal carriage were performed using a similar methodology in a close-time period but targeting different age-cohorts; the first study [33,34] found a 4% prevalence in adolescents from 18 to 24 years old; and the second one [33,34] found and overall of 6.5% in adolescents from 10 to 19 years old, with 7.6% in those from 14 to 19 years old.

#### 3.3.5. Vaccine Strategy and Uptake

Among countries that have implemented routine vaccination against *N. meningitidis,* Chile is the only one not targeting adolescents (Table 2). MCV-ACWY was implemented in toddlers in 2014 after a MenW outbreak, with a high uptake rate median of 95.6% (IQR = 92.0–97.0%) during the period 2014–2021. Recently, vaccination against MenB was introduced in NIP during the second semester of 2023, Chile being the first Latin American country adopting this strategy (coverage data are not available yet) (Table 2, and Appendix A). 

### 3.4. MD in Uruguay 

#### 3.4.1. Incidence

Median overall incidence during the period 2010–2021 was 0.47/100,000 (IQR 0.33–0.69). In contrast to Argentina, Brazil, and Chile, a sharp increase in incidence rates was observed in Uruguay from 2016 with a peak of 0.88/100,000 inhabitants in 2019. When data corresponding to years 2020 and 2021 were excluded, median incidence rate increased to 0.58/100,000 (IQR 0.40–0.73), with a percentage decrease of 72.6% in 2020 compared to the median in the period 2010–2019 (Table 1). 

#### 3.4.2. Serogroup Distribution

MenB was the predominant identified serogroup during each year of the period 2010–2019, while MenW had lower and relatively stable frequencies during the same period. In contrast to the other countries, databases from Uruguay report significant proportions of cases with no identification of serogroup, especially during the period 2017–2021. No cases due to MenA were reported during the period (Figure 3).

#### 3.4.3. Case Fatality Rates

Median CFR during the period 2010–2021 was 17.9% (IQR 8.5–21.3%), with a peak of 33% in 2021 (Figure 4).

#### 3.4.4. Carriage

No studies were performed in Uruguay.

#### 3.4.5. Vaccine Strategy and Uptake

Routine vaccination in the healthy population has not been introduced in Uruguay. 

## 4. Discussion

Meningococcal disease is a global threat that challenges the public health due to its dynamic and everchanging epidemiology with a high CFR and frequent sequelae. The cornerstone for *N. meningitidis* transmission is the pharyngeal carriage of adolescents, which generally occurs in an asymptomatic manner for a long time, being a dynamic process. WHO recommends implementing different health strategies to defeat meningitis by 2030, for which it is necessary to have evaluations adjusted to local epidemiological surveillance to establish the most appropriate recommendations for each country. Consequently, a sustained surveillance program of MD, including serogroups and age-specific distribution, is crucial to guide prevention strategies, even after the implementation of vaccination to monitor its impact on disease burden. In this context, efforts have been made in Latin America to improve surveillance systems [10,23,24,25,26,27,28,35,36], including a standardized definition for cases [37]. Countries that implemented meningococcal vaccines in NIP reduced the disease burden due to the vaccine serogroups, unlike Uruguay where an increase in the overall incidence of MD was observed, reaching the highest in the 2017–2019 period compared with the other three countries. Despite reductions in incidence, CFR remains high in all four countries (Appendix A).

It Is noteworthy that during 2020, a significant decrease in overall incidence of MD was seen in all countries, as has been described in other regions [38], probably due to COVID-19 containment policies and all non-pharmacological interventions implemented. 

Although MD occurs in all age groups, incidence in our study was highest in infants and young children, with no second peak in adolescents [26,29,39], Chile being the only country reporting a second peak in people older than 60 years old [36]. Despite the absence of a peak in adolescents and young adults in Brazil, an important number of MD cases occurred among these age cohorts. The switching in the predominance of serogroups from MenC in Brazil and MenW in Argentina and Chile to MenB occurred in three countries in our study, but we did not observe an increase in their incidence rates of MenB, which remained stable in Brazil [40] and Chile [10], while data from Argentina on incidence by serogroup are not available. Conversely, in Uruguay MenB has predominated across all of the analyzed period; however, incomplete data of serogroups are available in a considerable number of cases since 2017. 

Currently, effective vaccines, with proven direct protection, are available to protect against five of the six main disease-causing serogroups [36,41,42,43,44,45,46]. Moreover, when implementing vaccination strategies using MCV formulations, it is recommended to include adolescents regardless of their MD incidence due to the potential to influence the prevalence of carriage, further reducing transmission to non-vaccinated cohorts [6,17,18,47], as has been reported from the United Kingdom and the Netherlands [9,46,48] and which is the most likely explanation for the indirect effect. 

Different strategies for MCV were put in place among the countries in South America, but all have shown impact decreasing the MD incidence (Table 1). Notwithstanding the recommendation of including adolescents to achieve an indirect effect, only Argentina executed it since the beginning of its vaccination strategy; however, the median uptake has been poor across all age cohorts. In 2010, Brazil was the first country in the region that introduced MCV-C into NIP, using a 2 + 1 schedule, but despite good uptake in these cohorts, the lack of impact in non-vaccinated cohorts prompted the inclusion of adolescents in the NIP, first with MCV-C in 2017, followed by MCV-ACWY in 2020; then again, the uptake has been poor. Chile was the first country introducing a MCV-ACWY into the NIP in 2014, using a one-dose schedule at 12 months of age after a campaign from October 2012 to December 2013 targeting children from 9 months to 4 years of age, reaching high uptakes since its implementation, but at the moment an adolescent strategy has not been implemented yet and is considered to be carried out during 2024 [49]. 

There is consensus on the fact that to achieve high levels of direct and/or indirect protection, high uptakes of vaccination are required [50,51] for priming and booster doses. Efforts are necessary to clearly emphasize the need to implement strategies to improve vaccine uptake in all targeted cohorts. Adherence to vaccines during adolescence is a global issue [52,53], influenced by multiple factors such as the lack of programmatic preventive health-care visits at this age, weak clinician advisement about the importance of vaccination, and the lack of school entry requirements for vaccination of adolescents [54]. Therefore, a deep analysis inside each country should be performed to detect main causes of suboptimal vaccine uptake, especially in adolescents, considering that the indirect protection benefit relies on its coverage. Regarding Uruguay, at the moment a non-meningococcal vaccine program has been implemented, despite similarities in incidences, disease burden, and CFR with the other mentioned three countries. Further analysis improving serogroup identification after the pandemic will be necessary to discuss the epidemiological data and the potential vaccine introduction in Uruguay.

After the implementation of MCV, MenB has increased its frequency in Argentina, Brazil, and Chile, and it is the predominant serogroup in Uruguay. Two licensed MenB recombinant protein vaccines are available (4CMenB [55] and MenB-FHbp [56]) proving to be safe and immunogenic in most of population, but only 4CMenB is approved for use in infants and young children [12,42,57,58,59,60,61,62,63,64]. More recently, the evidence of real-world effectiveness of the routine use of 4CMenB in infants was published, showing its effectiveness not only against MenB [42,65,66,67] but also non-MenB cases [67]. Recently, Chile introduced 4CMenB in infants in a two-dose schedule during June 2023, being the first country in Latin American to implement it into the NIP, but other countries might consider it in the near future, depending on the evolution of the MD burden in the region. The costs and the lack of effect of these MenB vaccines on carriage will drive decisions on how to put them in place [68,69,70,71], targeting the recommendations to protect the age groups with the highest burden of disease as well as high-risk groups and outbreak control [72]. 

Based on these data and considering that in the South American countries included in this article, (1) the decrease in MD incidence rates in the pre-pandemic period was only observed in the countries that have implemented routine meningococcal vaccination programs, in contrast to Uruguay; (2) CFRs have remained unchanged and particularly high in Brazil, Chile, and Uruguay; and (3) though adolescents have lower MD incidence rates compared to young children, they have the highest rates of carriage; a routine vaccination MCV program targeting infants or/and toddlers must include adolescents, regardless of their incidence, to provide direct and indirect protection across all age groups as the ideal recommended program for MCV. 

This article is not free of limitations. Epidemiological data from the four included countries are not complete despite the existing surveillance systems, and the included countries may not be representative of the whole South American region. This highlights the need for continuous efforts to improve notification, diagnosis, and vaccination uptake in the region to reduce the burden of MD. 

## 5. Conclusions

A sustained surveillance program of MD, including serogroups and age-specific distribution, is crucial to guide prevention strategies after their implementation in the NIP to monitor their impact on disease burden. Experience after the implementation of MCV programs in South America was successful, reducing the burden of MD due to the vaccine serogroups. Nevertheless, it is necessary to enhance strategies to assure a high vaccine uptake across all ages and the inclusion of adolescents, which will be crucial in the postpandemic period to maintain the protection of the population. 

## Figures and Tables

**Figure 1 vaccines-11-01841-f001:**
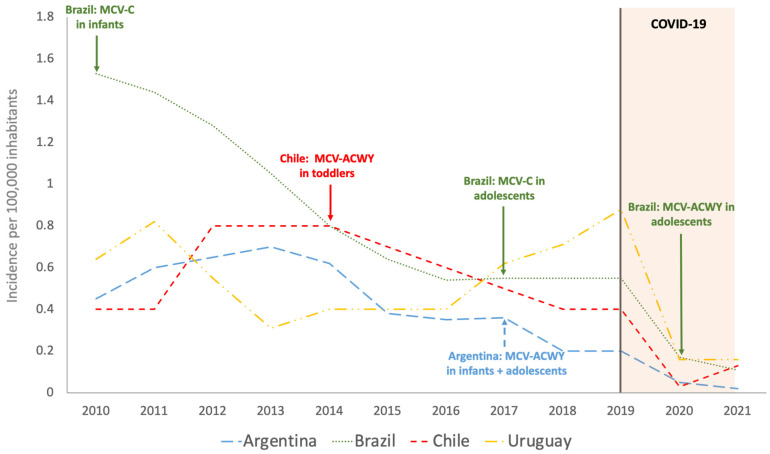
Overall incidence of meningococcal disease in Argentina, Brazil, Chile, and Uruguay during the period 2010–2021. MCV-C: meningococcal conjugate vaccine against serogroup C; MCV-ACWY: meningococcal conjugate vaccine against serogroups A, C, W, and Y. The year of introduction of meningococcal conjugated vaccine (MCV) into National Immunization Programs at determined ages is pointed out for each country.

**Figure 2 vaccines-11-01841-f002:**
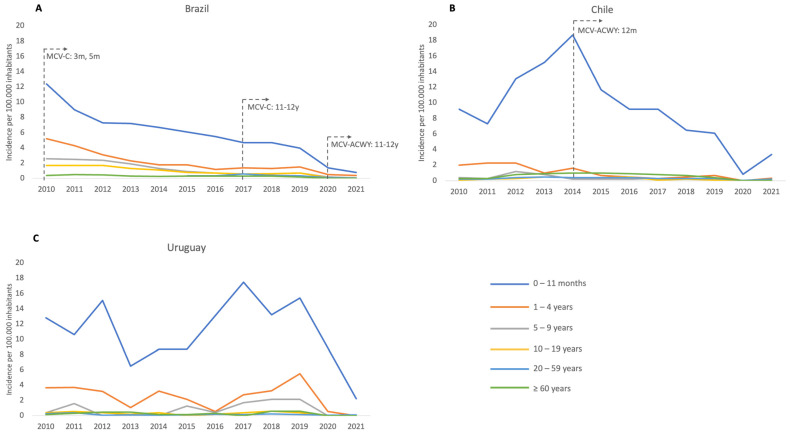
Incidence of meningococcal disease by age group during the period 2010–2021 in (**A**) Brazil, (**B**) Chile, and (**C**) Uruguay. The year of introduction of meningococcal conjugated vaccine (MCV) into NIP at determined ages is pointed out for each country; m = months of age and y = years of age; MCV-C: meningococcal conjugate vaccine against serogroup C MCV-ACWY: meningococcal conjugate vaccine against serogroups A, C, W, and Y.

**Figure 3 vaccines-11-01841-f003:**
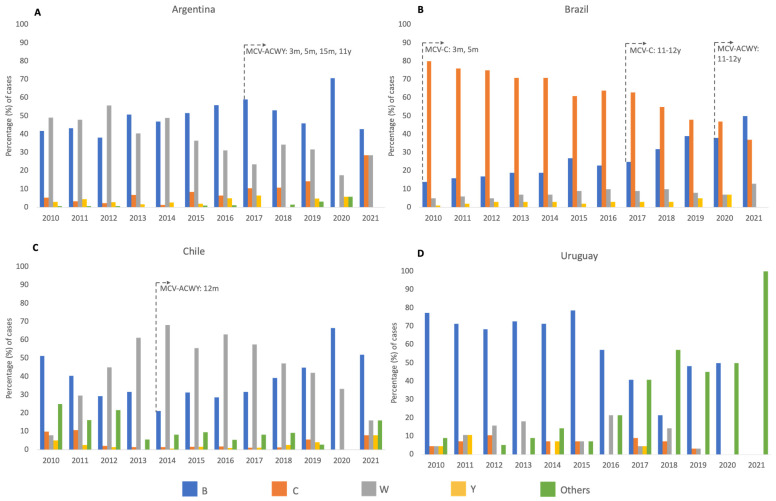
Percentage distribution of serogroups in cases of meningococcal disease by year in Argentina (**A**), Brazil (**B**), Chile (**C**), and Uruguay (**D**), 2010–2021. The year of introduction of meningococcal conjugated vaccine (MCV) into National Immunization Programs at determined ages is pointed out for each country; m = months of age, and y = years of age. “Others” include non-BCWY serogroups, non-serogroupable, and unknowns. No cases due to MenA were reported in Argentina, Brazil, Chile, or Uruguay during the period.

**Figure 4 vaccines-11-01841-f004:**
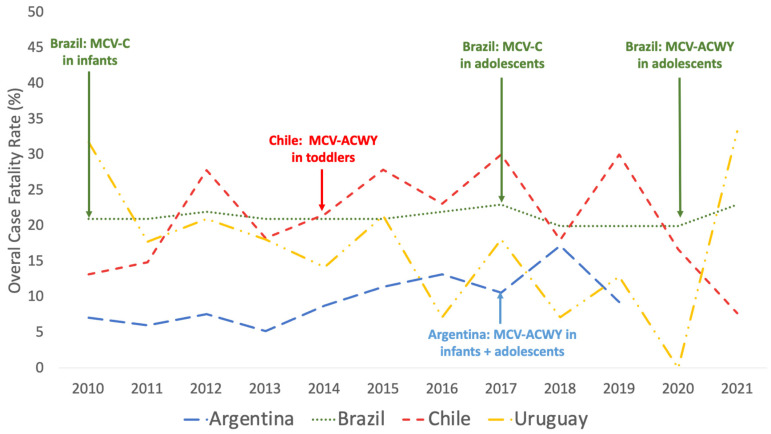
Case Fatality Rate of Meningococcal Disease during the period 2010–2021 in Argentina, Brazil, Chile, and Uruguay. Data from Argentina are available until 2019. The year of introduction of meningococcal conjugated vaccine (MCV) into National Immunization Programs at determined ages is pointed out for each country.

**Table 1 vaccines-11-01841-t001:** Epidemiological and vaccination characteristics of meningococcal disease in Argentina, Brazil, Chile, and Uruguay during the period 2010–2021.

	Argentina	Brazil	Chile	Uruguay
Median Incidence (IQR)/100,000 inhabitants				
2010–2021	0.37 (0.20–0.61)	0.59 (0.54–1.22)	0.45 (0.4–0.77)	0.47 (0.33–0.69)
2010–2019	0.41 (0.31–0.62)	0.72 (0.55–1.32)	0.55 (0.40–0.80)	0.58 (0.40–0.73)
Incidence reduction during pandemic ^a^	87.9%	76.3%	94.5%	72.6%
Disease burden	Infants Children 1–4 yoa	Infants Children 1–4 yoa and young adults	Infants Children 1–4 yoa and >60 yoa	Infants Children 1–4 yoa
Serogroup (predominance)	W + B	C + B	W + B	B
Carriage (%) (adolescents)	9.4	9.9–12.5	6.5	N/A
Case fatality rate (Median %)	9	21	19.9	17.9

^a^ Pandemic impact was estimated comparing periods 2010–2019 versus 2020: 100 × ((Median 2010–2019) − 2020)/(Median 2010–2019)).

**Table 2 vaccines-11-01841-t002:** Vaccination strategies in Argentina, Brazil, and Chile during the period 2010–2021.

	Argentina	Brazil		Chile	
Vaccine in NIP ^a^	MCV-ACWY	MCV-C	MCV-ACWY	MCV-ACWY	4CMenB
Age of vaccination					
Infants	3 and 5 moa	3 and 5 moa	N/A	N/A	2 and 4 moa
Toddlers	15 moa	12–15 moa	N/A	12 moa	N/A
Adolescents	11 yoa	N/A	11–12 yoa	N/A	N/A
Year of implementation	2017	2010/2017 ^c^	2020	2014	2023
Vaccine uptake (%) ^b^					
Infants	66.6	91	N/A	N/A	N/A ^e^
Toddlers	54.7	84.5	N/A	95.6	N/A
Adolescents	47.5	53 ^d^		N/A	N/A

^a^ NIP: National immunization program. ^b^ Vaccine uptake was estimated as median percentage. ^c^ Brazil introduced MCV-C for infants and toddlers in 2010. The adolescents were included in 2017 first using MCV-C, followed by MCV-ACWY in 2020. ^d^ Includes coverage of both MCV-C and MCV-ACWY during the period. ^e^ Data about the uptake of 4CMenB in infants are not available due to its recent implementation. N/A = not applicable (grayed out for better visualization of available data).

## Data Availability

Epidemiological data from Chile was extracted from publicly datasets, which can be found here: http://epi.minsal.cl/enfermedad-meningococcica-bases-de-dato, https://www.ispch.cl/sites/default/files/Bolet%C3%ADnMeningococo-06072020A.pdf, https://vacunas.minsal.cl/wp-content/uploads/2022/04/Informe-de-Coberturas_2021_enero_diciembre.pdf. Epidemiological data from Brazil was extracted from publicly datasets, which can be found here: http://tabnet.datasus.gov.br/cgi/tabcgi.exe?sinannet/cnv/meninbr.def (all accessed on 5 May 2023). Restrictions apply to the availability of data from Argentina and Uruguay; data were requested from the MoH of each country, and is available with their respective permissions.

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
