# Peer review of "Epidemiology of Meningococcal Disease in Four South American Countries and Rationale of Vaccination in Adolescents from the Region: Position Paper of the Latin American Society of Pediatric Infectious Diseases (SLIPE)"

_vaccines, 2023, doi:10.3390/vaccines11121841_

Round 1

Reviewer 1 Report

Comments and Suggestions for Authors

This paper had reviewed the epidemiology of MD in four South American countries with well-established surveillance systems (Argentina, Brazil, Chile, and Uruguay) over the period 2010 - 2021. Epidemiological data were obtained from the Ministry of Health (MoH) databases of four countries. This article provided a systematic descriptive analysis of reported cases of MD in four countries. The distribution of MD incidence, distribution of specific serogroups, mortality, carriage, and vaccine strategy and vaccination rates were described and analysed for each of the four countries.

The paper was well written, which can provide basic information on the prevalence of MD in these four countries for subsequent prevention and vaccine development. However, there were some problems, which must be solved before it is considered for publication.

1.

In page 1-ABSTRACT

25-26we analyze the epidemiology of MD and vaccination experience from these four South Amer- 25 ican countries.”

No indication of what the purpose of the analysis in this paper is.

2.

In page 2-INTRODUCTION

Relevant research background needs to be supplemented in INTRODUCTION,almost as if there was no mention of what previous work on the subject has been like. And you should cite all papers you use properly.

3In page 4-RESULTS

The symbols in Table 1 were too complex, so its recommended to use simple and clear symbols, such as a, b, c, d etc. The annotations for Table 1 should be in the order of the variables in the table. In addition, you could reduce some of the symbols, such as "& Overall incidence is expressed as a median per 100,000 inhabitants", which can be expressed in the rowname Incidence ( per 100,000) .

4.In page 6-RESULTS

Figure 3 consists of four pictures named A, B, C, and D. However, the picture D was labelled in a wrong position.

5.Suggested improvements to figure

It is recommended that the incidence of MD, case fatality rate of MD, and vaccination coverage of MD in each country be placed on a single graph to facilitate the observation of changes in the incidence of disease in each country as a result of vaccine application. Authors could put this single figure in the mamuscript or supplementary file. 

Author Response

We thank to Reviewer 1 your useful suggestions. 

In page 1-ABSTRACT

25-26:“we analyze the epidemiology of MD and vaccination experience from these four South Amer- 25 ican countries.”

No indication of what the purpose of the analysis in this paper is.: The aim of the analysis was added to the abstract (line 26): “to identify needs and plans to improve the current vaccination programs”

2.

In page 2-INTRODUCTION

Relevant research background needs to be supplemented in INTRODUCTION, almost as if there was no mention of what previous work on the subject has been like. And you should cite all papers you use properly. A brief comment about research background (recent publications about epidemiology of MD in latin American countries) and the gap of knowledge (no publications comparing data of different countries) has been added to the introduction section (lines 80-85)

3In page 4-RESULTS

The symbols in Table 1 were too complex, so it’s recommended to use simple and clear symbols, such as a, b, c, d etc. The annotations for Table 1 should be in the order of the variables in the table. In addition, you could reduce some of the symbols, such as "& Overall incidence is expressed as a median per 100,000 inhabitants", which can be expressed in the rowname Incidence ( per 100,000) . Table 1 was modified according to your suggestions. Also, data about vaccination strategies was moved to a new Table (Table 2).

4.In page 6-RESULTS

Figure 3 consists of four pictures named A, B, C, and D. However, the picture D was labelled in a wrong position. Corrected. Also, Figure 3 was modified to a grouped column bar graphic to a better visualization of serogroup distribution and changes over time.

5.Suggested improvements to figure

It is recommended that the incidence of MD, case fatality rate of MD, and vaccination coverage of MD in each country be placed on a single graph to facilitate the observation of changes in the incidence of disease in each country as a result of vaccine application. Authors could put this single figure in the mamuscript or supplementary file. Supplementary figure 1 including overall incidence rates, case fatality rates and vaccination coverage was added according to your suggestion. Also, figure 5 (vaccination coverage) was removed, as the main results about this are included in the main text and in table 2.

Reviewer 2 Report

Comments and Suggestions for Authors

Review of ‘Epidemiology of Meningococcal disease in four South Americancountries, and rationale of vaccination in adolescents from the region:Position paper of the Latin American Society of Pediatric Infectious Diseases

 This article describes meningococcal epidemiology in four South American countries since 2010. The data includes overall incidence, serogroup distribution, CFR and data on vaccine schedules/uptake. The article is well written and would be of interest to epidemiologists, microbiologists and vaccine policy makers. I believe changes could be made to improve the paper and make the findings easier to follow by the reader.

 Major comments

Results: I would suggest reformatting the results so that there are separate sections for each country in turn. This would make it easier for the reader to understand the situation in each country. I find the current format, where the authors keep switching back and forth between different countries makes it hard to understand each situation.

Table 1: I suggest the information on the different vaccine schedules in each country should be moved to a separate table. This might allow for a better format to describe the situation and a comments section could be included to prevent the need for all the footnotes

Figure 3: I recommend changing the figure to stacked columns rather than % columns. This would help the reader understand the change in disease of each serogroup. I would also add the CFR lines from figure 4 on to Figure 3 (using a secondary axis) and remove Fig 4. This would help readers to see the underlying case numbers, serogroup distribution and the effect this has on the CFR over time all on one graph.

Figure 5: Not sure what this data adds as its very limited (a lot of gaps). Most of the data is described in the text anyway so would consider removing fig 5 completely.

Discussion: a lot of the discussion seems to be repeating the results, especially when quoting values from the results. I suggest revising the discussion and making the discussion more concise with less numbers and more interpretation/conclusions.

Minor comments

Introduction, second paragraph, line two: not sure if “deserve” is the correct term to use here.

Lines graphs: I suggest you distinguish between different lines by changing the lines (e.g. dotted lines, dots and dashes, solid lines) rather than colour. This makes it easier to read when printed in black and white.

Line 288: “with Chile..” instead of “being Chile…”?

Lines 289 and 296: Not sure if relevant is the correct word.

Line 295: What do you mean by ‘non-accurate’. If it’s inaccurate, should it be used?

Line 315: Lack not lacking

Comments on the Quality of English Language

The paper is very well written, although I have queried a couple of word choices in my minor comments.

Author Response

Thanks Reviewer 2 for your useful suggestions. 

Results: I would suggest reformatting the results so that there are separate sections for each country in turn. This would make it easier for the reader to understand the situation in each country. I find the current format, where the authors keep switching back and forth between different countries makes it hard to understand each situation. We have accepted your suggestion

Table 1: I suggest the information on the different vaccine schedules in each country should be moved to a separate table. This might allow for a better format to describe the situation and a comments section could be included to prevent the need for all the footnotes. We have accepted your suggestion, and moved data about vaccination schedules and coverage to table 2.

Figure 3: I recommend changing the figure to stacked columns rather than % columns. This would help the reader understand the change in disease of each serogroup. I would also add the CFR lines from figure 4 on to Figure 3 (using a secondary axis) and remove Fig 4. This would help readers to see the underlying case numbers, serogroup distribution and the effect this has on the CFR over time all on one graph. Considering that the original figure 3 was indeed a stacked column graph, we think the reviewer meant to change it to a grouped column graph. We consider that the new version offers a better visualization of serogroup distribution and changes over time than the previous version. However, we prefer not to add the CFR lines to this figure, to avoid an excess of information, and because we do not find a clear link between serogroup distribution and CFR.

Figure 5: Not sure what this data adds as its very limited (a lot of gaps). Most of the data is described in the text anyway so would consider removing fig 5 completely. Figure 5 was removed, and data about vaccine coverage in different age groups was added in supplementary figure 1 (together with overall incidence and CFR according to another reviewer’s suggestion).

Discussion: a lot of the discussion seems to be repeating the results, especially when quoting values from the results. I suggest revising the discussion and making the discussion more concise with less numbers and more interpretation/conclusions. We have accepted your suggestion, and eliminated  unnecessary numerical data

Minor comments

Introduction, second paragraph, line two: not sure if “deserve” is the correct term to use here. It has been replaced by “require”

Lines graphs: I suggest you distinguish between different lines by changing the lines (e.g. dotted lines, dots and dashes, solid lines) rather than colour. This makes it easier to read when printed in black and white. We have accepted your suggestion and changes the lines’ structure in figure 1 and figure 4, maintaining the colours. However, in figure 2 we did not change the lines’ structure because lower lines would not be easily distinguished (they are overlapping).

Line 288: “with Chile..” instead of “being Chile…”? corrected

Lines 289 and 296: Not sure if relevant is the correct word.: They have been changed to “important” and “considerable” respectively.

Line 295: What do you mean by ‘non-accurate’. If it’s inaccurate, should it be used? It was changed to “incomplete”

Line 315: Lack not lacking: Corrected

Round 2

Reviewer 1 Report

Comments and Suggestions for Authors

1. 

In table1,

Incidence per 100,000 inhabitants.

Median (IQR)

suggested revision to “Mean Incidence  (IQR) / 100,000 inhabitants

The first two variable rows in table1 are not horizontally aligned.

Author Response

Thanks to Reviewer 1 for his/her suggestions: 

In table1, “Incidence per 100,000 inhabitants. Median (IQR)” suggested revision to “Mean Incidence  (IQR) / 100,000 inhabitants”: We have accepted your suggestion, and correct it to "Median Incidence (IQR)/ 100,000 inhabitants"

The first two variable rows in table1 are not horizontally aligned: Corrected